# Activated Carbon and Carbon Quantum Dots/Titanium Dioxide Composite Based on Waste Rice Noodles: Simultaneous Synthesis and Application in Water Pollution Control

**DOI:** 10.3390/nano12030472

**Published:** 2022-01-29

**Authors:** Xinyan Jin, Ruijie Che, Jie Yang, Yan Liu, Xinbao Chen, Yunge Jiang, Jiaqi Liang, Shuoping Chen, Heping Su

**Affiliations:** 1College of Materials Science and Engineering, Guilin University of Technology, Guilin 541004, China; 2120200261@glut.edu.cn (X.J.); 2120210253@glut.edu.cn (R.C.); 2120180167@glut.edu.cn (J.Y.); 2120200279@glut.edu.cn (Y.L.); 3172042041419@glut.edu.cn (X.C.); 3192042041221@glut.edu.cn (Y.J.); 3192042041306@glut.edu.cn (J.L.); 2College of Science, Guilin University of Technology, Guilin 541004, China

**Keywords:** waste rice noodle, activated carbon carbon, quantum dots/titanium dioxide composite, water pollution control

## Abstract

To achieve the full utilization of waste rice noodle (WRN) without secondary pollution, activated carbon (AC) and carbon quantum dots/titanium dioxide (CQDs/TiO_2_) composite were simultaneously synthesized by using WRN as raw material. Both of the two materials showed potential applications in water pollution control. The AC based on WRN displayed a porous spherical micro-morphology, which could absorb heavy metal elements like Pb(II) and Cr(VI) efficiently, with a maximum equilibrium uptake of 12.08 mg·g^−1^ for Pb(II) and 9.36 mg·g^−1^ for Cr(VI), respectively. The adsorption of the resulted AC could match the Freundlich adsorption isotherm and the pseudo-second-order kinetics mode. On the other hand, the CQDs/TiO_2_ composite based on WRN displayed a high efficient photocatalytic degradation effect on various water-soluble dyes such as methylene blue, malachite green, methyl violet, basic fuchsin, and rhodamine B under visible light irradiation, which showed better photocatalytic performance than commercial TiO_2_. The introduction of CQDs based on WRN to TiO_2_ could result in efficient electron-hole pair separation and enable more photogenerated electrons to reduce O_2_ and more photogenerated holes to oxidize H_2_O or OH^−^, which could cause stronger abilities in producing O_2_^·−^ and ·OH radical and better photocatalytic activity.

## 1. Introduction

Cooking waste can cause potential pollution to the soil and water environment and provide a significant challenge to urban governance [1,2,3]. Nowadays, commercial treatments for cooking waste recycling are mainly anaerobic digestion [4,5,6], aerobic composting [7,8], landfill [9,10,11], incineration [12], and forage making [7]. Although cooking waste can be treated through the above strategy with large-scale industrialization, there are some fatal shortcomings, such as the mass occupation of land, high investment for equipment, low product profit margin, easy production of secondary pollution like greenhouse gases, and waste leachate [1,13].

In recent years, hydrothermal carbonization has proved to be a new and effective method for cooking waste treatment [14,15,16], whose process is simple, environment friendly, requires mild conditions, results in near-zero emissions, and may provide functional carbon materials such as activated carbon (AC) [17] or fluorescent carbon quantum dots (CQDs) [18]. However, due to complicated compositions, the intermediate products of hydrothermal carbonization often display different physical chemical characters, which produce great difficulties for subsequent treatment. Thus, it is difficult to achieve the full utilization of cooking waste, and secondary pollution like waste water and residue is unavoidable.

The AC with a porous structure can effectively absorb various kinds of metal ions and is widely used in the treatment of wastewater [19]. For cost control and resource recycling, a wide variety of AC adsorbents derived from a natural products or organic waste have been reported in recent years, whose sources include coconut shell [20,21], pecan shell [22], peanut shell [23], date pit [24], hazelnut husk [25], sagowaste [26], paper mill sludge [27], polygonum [28], crassipes root [29], and so on. On the other hand, nano titanium dioxide (TiO_2_) is a commercial photocatalyst used in water and air pollution control [30,31,32]. However, pure TiO_2_ displays a wide band gap whose photogenerated electrons and holes are easy to combine again. Moreover, it has a low absorption and utilization rate of visible light whose wavelength is more than 380 nm, which limits its application in visible light catalysis. As a new type of carbon nano-material, CQDs have many advantages, such as excellent stability against photobleaching, good water solubility, high biocompatibility, and relatively low toxicity [33,34,35] and can also be used as photoactive materials to generate reactive oxygen species under light radiation [36]. To date, CQDs-enabled photocatalysts are regarded as among the most efficient technologies to degrade organic pollutants in water [37,38,39]; for the introduction of CQDs to inorganic photocatalyst like TiO_2_ can effectively enhance energy absorption, promote electronic transition, produce more electron-hole pairs, and reduce the recombination of electron-hole pairs [40,41,42]. However, the current commercial CQDs are very expensive, which hinders the commercialization of the CQDs/TiO_2_ composite.

Rice noodle is a traditional daily diet of local people in Guilin city, China. Therefore, a large amount of waste rice noodles (WRN) is produced in this city, with an output of over 70 tons per day. However, due to the underdeveloped economy of Guilin, the infrastructure, policy support, and capital investment in the recycling of catering waste are relatively limited. Most of the WRN is used as raw materials for low-degree fermentation feed, and the other part is directly discarded, which causes potential and long-term pollution to the soil and water environment. Therefore, it is necessary to explore an effective way to recycle WRN. Starch is the main organic component in WRN, which is easy to carbonize under hydrothermal conditions, so it has the potential to prepare functional carbon materials. In our previous works, we treated WRN by a hydrothermal method to obtain carbon spheres and CQDs and then blended the resulting carbon materials with polylactic acid (PLA), resulting in C/PLA and CQDs/PLA 3D printing composites, respectively [43,44]. Herein, in order to achieve the full utilization of WRN and obtain low-cost carbon materials for water pollution control, we design a new method for co-producing AC and CQDs/TiO_2_ composite with WRN as raw materials (See Figure 1). More specifically, WRN was treated under hydrothermal conditions to form a mixture of precipitate (hydrothermal carbon powder, HTC powder) and liquid (CQDs solution), and then the HTC powder was produced as AC, while the CQDs solution could be used to synthesize CQDs/TiO_2_ composite simultaneously. Furthermore, the obtained AC could absorb heavy metal elements, while the CQDs/TiO_2_ composite displayed a high efficient photocatalytic degradation effect on water-soluble dyes, which could together form a wastewater treatment system.

## 2. Experiment Section

### 2.1. Materials and Methods

The waste rice noodle (WRN, main organic constituent: starch 21.36 g/100 g; protein 1.91 g/100 g; fat 0.4 g/100 g) was collected from the canteen of the Guilin University of Technology in Guilin, China. The sodium carboxymethyl cellulose (CMC-Na), potassium hydroxide (KOH), hydrochloric acid (HCl), potassium dichromate (K_2_Cr_2_O_7_), lead nitrate (Pb(NO_3_)_2_), nano-TiO_2_ (P25 grade), methylene blue, malachite green, methyl violet, basic fuchsin, and rhodamine B were all of the analytical reagent grade and purchased from Macklin Reagent (Shanghai, China) without further purification.

### 2.2. Hydrothermal Treatment of WRN

A total of 100 g of WRN was ground to a smooth paste in a mortar and then mixed with 200 g deionized water. The resulted mixture was transferred to a 500 mL sealed Teflon-lined autoclave (Kemi Instrument, Anhui, China) and then heated at 200 °C for 10 h. A kind of brown solution with black–gray precipitation was obtained. The solid part (HTC powder) can be further processed to obtain activated carbon, while the liquid part (CQDs solution) could be used to obtain CQDs/TiO_2_ composite. For 100 g of WRN, 13.7 g hydrothermal carbon powder and about 200 mL CQDs solution could be obtained.

### 2.3. Synthesis of AC Based on WRN

Under optimized conditions, 13.7 g HTC powder prepared in the previous step was mixed with 27.4 mL 1% CMC-Na solution and 27.4 g KOH. The mixture was stirred to form a black plastic paste. It could be formed by die or handcraft to obtain a billet with an appropriate shape. After drying for 24 h at 60 °C, the billet of AC was sent to a tubular furnace protected by nitrogen and activated at 800 °C for 90 min. After activation, the resulting black product was taken out, and the excess alkali in AC was washed off by adding 10% HCl. The product was then washed to neutral by hot distilled water for four times and dried at 60 °C for 24 h. In this way, black AC based on WRN was obtained, which could be prepared into powder, particle, or other specific shapes. The conversion rate of AC was 31.6% based on HTC powder, and 4.3 g activated carbon could be obtained with 100 g WRN as raw material. The optimization of the experimental conditions of AC is shown in Appendix A.

### 2.4. Synthesis of CQDs and CQDs/TiO_2_ Composite Based on WRN

On the other hand, the liquid part (CQDs solution) was filtered and dialyzed in dark for 24 h, resulting in a fluorescent CQDs water solution, which could be directly used for Fe^3+^ detection (see Appendix A). However, for the synthesis of CQDs/TiO_2_ composite, dialyzation was not a necessary step. After filtration, 200 mL CQDs solution was directly mixed with 2.5 g TiO_2_ powder, and the mixture was stirred at room temperature for 30 min to obtain a uniform suspension; then the suspension was transferred to an oven and kept at 85 °C for 3 h. The CQDs/TiO_2_ composite was centrifuged, washed with distilled water for three times, and finally dried overnight at 60 °C in a vacuum environment, which gave a dark brown CQDs/TiO_2_ composite with a carbon content of 15.24%. Thus, 2.9 g CQDs/TiO_2_ composite could be obtained with 100 g WRN as raw material. The optimization of the experimental conditions of CQDs/TiO_2_ composite is shown in Appendix A.

### 2.5. General Characterization

The infrared (IR) spectra of the resulting products were recorded as KBr pellets at a range of 400–4000 cm^−1^ on a Nicolet 5700 FT-IR spectrometer (Thermofisher, Waltham, MA, USA) with a spectral resolution of 4.00 cm^−1^. The powder X-ray diffraction (PXRD) patterns were obtained with an X ‘pert PRO X-ray diffractometer (Panalytical, Malvern, Worcestershire, UK) with Cu Ka radiation (λ = 0.15418 Å) at 40 kV and 40 mA and a scan speed of 5° min^−1^ (2θ). The scanning electron microscope (SEM) images and energy-dispersive x-ray (EDX) spectra of AC were obtained by using an S-4800 field emission scanning electron microscope (Hitachi, Chiyoda, Tokyo, Japan) with an accelerating voltage of 20 kV. The morphology of CQDs and CQDs/TiO_2_ composite were characterized by using JEM-2100F field emission transmission electron microscope (TEM, JEOL, Akishima, Tokyo, Japan) with an accelerating voltage of 200 kV. The photoluminescence spectra of pure TiO_2_ and CQDs/TiO_2_ composite were examined with a Cary Eclipse fluorescence spectrophotometer (Varian, Palo Alto, CA, USA) with a xenon lamp as the excitation source in the range of 350 to 650 nm and an excitation wavelength of 325 nm. The BET-specific surface tests of resulted in AC and other control samples were characterized by using a TriStar II 3020 surface area analyzer (Micromeritics, Atlanta, GA, USA) under a nitrogen atmosphere with an initial temperature of 150 °C and a heating rate of 10 °C/s. The X-ray photoelectron spectroscopy (XPS) of AC and CQDs/TiO_2_ composite were carried out with an ESCALAB 250Xi X-ray photoelectron spectrometer (Thermofisher, Waltham, MA, USA) with an Al Kα x-ray as the stimulated source. The UV-VIS absorption spectra of CQDs/TiO_2_ composite were measured by an UV3100 UV-VIS-NIR spectrophotometer (Shimadzu, Chiyoda, Tokyo, Japan) in diffuse reflection mode, using BaSO_4_ as reference. The electrochemical properties for pure TiO_2_ and CQDs/TiO_2_ composite were investigated in a three-electrode system with a platinum network counter electrode and an Ag/AgCl (saturated KCl) reference electrode, using a concentration of 0.1 M Na_2_SO_4_ aqueous electrolyte at 25 °C. The transient photocurrent response was performed on an electrochemical analyzer using 0.5 V bias voltage under light irradiation with a 300 W Xe lamp as the light source. The radical production abilities of pure TiO_2_ and CQDs/TiO_2_ composite were investigated by electron paramagnetic resonance (EPR) with a EMXplus X-band electron paramagnetic resonance spectrometer (Bruker, Karlsruhe, Baden-Württemberg, Germany) using dimethyl pyridine N-oxide (DMPO) as spin trapping agent. The elemental analysis data (C, H, N) of AC and other control samples were obtained with a 240B elemental analyzer (PerkinElmer, Wellesley, MA, USA). The determination of the points of zero charge (PZC) of the resulted samples were carried out as follows [45]: 10 mL of 0.01 M NaCl solution was placed in a closed centrifugal tube. The pH was adjusted to a value between 2 and 11 by adding HCl or NaOH solutions. Then, 0.05 g sample was added, and the final pH was measured after 10 h under agitation at a sample. The point of zero charges (PZC) was obtained from the intersection of the pH_final_ vs. pH_initial_ curve of the test sample and that of the blank sample.

### 2.6. Measurement of Adsorption Performance of AC Based on WRN

Pb(II) and Cr(VI) were used to test the adsorption performance of obtained AC to a heavy metal element. Before the experiment, both Pb(II) and Cr(VI) were prepared as aqueous solutions with Pb(NO_3_)_2_ and K_2_Cr_2_O_7_ as element sources, whose pH were both adjusted to 5. Then, the heavy metal solution was mixed with AC with an adsorbent dosage of 7 g·L^−1^ in a crystallizing dish with continuous magnetic stirring at 25 °C. After absorption, the resulting solution was centrifugated, and the concentration of Cr(VI) or Pb(II) was tested with diphenylcarbazide spectrophotometry or dithizone spectrophotometry, respectively. The adsorption capacity of AC to Cr(VI) or Pb(II) was calculated according the spectrophotometry results by the following equation:(1)$$q_{e}=\frac{\left(C_{0}-C_{e}\right)}{AD}$$

Herein, *q_e_* is the equilibrium adsorption capacity of adsorbent in mmol·g^−1^; *C*_0_ is the initial concentration of the heavy metal ions in mg·L^−1^; *C_e_* is the equilibrium concentration of the heavy metal ions after adsorption in mg·L^−1^; *A* is the atomic weight of heavy metal in g/mol, and *D* is the adsorbent dosage in g·L^−1^.

The adsorption equilibrium was established within 12 h for both Cr(VI) or Pb(II) adsorption, with the initial concentration of heavy metal element ranging from 20–100 mg·L^−1^.

The Langmuir and Freundlich adsorption isotherms were applied to describe the adsorption of AC to Cr(VI) or Pb(II) at room temperature [46]. The Langmuir equations are as follows:(2)$$q_{e}=\frac{q_{m}bC_{e}}{1+bC_{e}}\;(\mathrm{nonlinear}\;\mathrm{form})\;\mathrm{or}\;\frac{C_{e}}{q_{e}}=\frac{1}{q_{m}^{b}}+\frac{1}{q_{m}}C_{e}\;(\mathrm{linear}\;\mathrm{form})$$
where *q_m_* is the maximum amount of heavy metal uptake in mmol·g^−1^; *b* is the constant that refers to the bonding energy of adsorption related to free energy and net enthalpy in L·mg^−1^.

The Freundlich isotherm is given as:(3)$$q_{e}=K_{f}C_{e}^{1/n}\;(\mathrm{nonlinear}\;\mathrm{form})\;\mathrm{or}\;\log q_{e}=\log K_{f}+\frac{1}{n}\log C_{e}(\mathrm{linear}\;\mathrm{form})$$
where *K_f_* is the constant related to the adsorption capacity of the adsorbent in mmol·g^−1^, and 1/*n* is the intensity of the adsorption constant.

The regression analysis was carried out by OriginPro software (OriginPro 2019, Originlab, Northampton, MA, USA) in order to predict the parameters.

The adsorption dynamics was tested with an initial concentration of 25 mg/L for both Cr(VI) or Pb(II) adsorption with a different adsorption time ranged from 10 to 120 min, and the results were fitted by pseudo-first-order or pseudo-second-order mode [47]. The pseudo-first-order equation was used as:(4)$$\log(q_{e}-q_{t})=\log q_{e}-\frac{K_{1}}{2.303}t$$
where *q_e_* and *q_t_* are the heavy metal uptake at equilibrium and time *t*, respectively, and *K*_1_ is the constant of first-order adsorption in min^−1^. 

The pseudo-second-order equation is given as:(5)$$\frac{t}{q_{t}}=\frac{1}{K_{2}q_{e}^{2}}+\frac{t}{q_{e}}$$
where *K*_2_ is the rate constant of second-order adsorption in g mmol^−1^ min^−1^.

The regression analysis was also carried out by OriginPro software to predict the parameters. 

### 2.7. Measurement of Photocatalytic Performance of CQDs/TiO_2_ Composite Based on WRN 

Photo-catalytic performance was tested by the photocatalytic degradation of dyes under visible light. First, a 50 mL dye solution with a concentration of 20 mg/L and 0.2 g CQDs/TiO_2_ composite was mixed in a crystallization dish with magnetic stirring and then placed under a 20 W 405 nm purple light (or xenon lamp) with an illuminance of 8 × 10^4^ LUX. Such samples were prepared for several repeat experiments. After a certain time, one of the crystallizing dishes was taken out. The resulted solution was centrifugated, and its dye concentration was tested by UV-VIS spectrophotometry. The photocatalytic rate of CQDs/TiO_2_ composite was calculated according to the spectrophotometry results.

The photocatalytic degradation of organic pollutants at the liquid–solid interface can be described by the Langmuir Hinshelwood model [48], in which the integral form is
(6)$$t=(\frac{1}{K_{r}K})\ln(\frac{C_{0}}{C})+\frac{C_{0}-C}{K_{r}}$$
where *t* refers to the irradiation time; *C*_0_ is the initial concentration of organic dye, and *C* is the concentration of pollutant at time *t*; *K* is the equilibrium constant for the organic dye adsorption on the catalyst, and *K_r_* reflects the limiting rate of the reaction at the maximum coverage under the given conditions.

At a low initial concentration of organic dye, the second term in Equation (6) becomes insignificant, and hence, it can be neglected. Thus, the kinetic fitting of photocatalytic degradation in this paper used the following equation:(7)$$\ln(\frac{C_{0}}{C})=K_{r}Kt=K_{app}t$$
where *K_app_* is the apparent constant in min^−1^ used as the basic kinetic parameter for different photocatalysts.

## 3. Result and Discussion

### 3.1. Structural Characterization of AC Based on WRN

As shown in Table 1, the HTC powder was a partially carbonized intermediate product with about 51 wt% carbon content. Due to the high starch content of WRN, the majority of the particles of HTC showed a spherical morphology with a smooth surface (Appendix A). Thus, the HTC displayed a very low surface area and could not be used for heavy metal adsorption directly. However, the HTC could generate thermal decomposition over 400 °C (Appendix A). Thus, by activation at 800 °C with KOH, AC for heavy metal adsorption could be obtained. Compared with HTC, the resulting AC based on WRN showed a higher graphitization degree, in which the broad peaks at about 25 and 43° represented the (002) and (100) crystal planes of graphite, respectively. The AC based on WRN showed the morphology of porous sphere with a dimension of 10–20 μm (see Figure 2b), which displayed a high BET-specific surface area of 1250.4 m^2^·g^−1^ and an average pore size of 2.48 nm, as well as a high adsorption value to methylene blue of 321 mg·g^−1^ and an iodine value of 961 mg·g^−1^. In addition, the AC net exhibited a type II isotherm in nitrogen adsorption and an H_2_/H_4_ hysteresis loop, which was a complex product composed of microporous and mesoporous structures (Table 1, Figure 2c,d). The IR spectrum revealed that there was a hydroxyl group in the surface of the resultant AC, in which the vibrations of σ_O-H_ and σ_C-O_ were located at 3441 cm^−1^ and 1039 cm^−1^, respectively, while the stretch of C=C in aromatic rings was located at 1636 cm^−1^ (Figure 2e). The XPS spectra showed the existence of C and O elements in the resulting AC (see Figure 2f). In the high-resolution spectrum of C 1 s, the characteristic peak at 284.28 eV, 286.01 eV, and 288.37 eV could be attributed to C–C bond, C–O bond, and C=C bond, respectively (see Figure 2g). The high-resolution spectrum of O 1 s showed the existence of C–O and O–H bonds, whose characteristic signals were located at 531.49 eV and 532.76 eV, respectively (see Figure 2h and Table 2).

### 3.2. Adsorption Performance to Heavy Metals of AC Based on WRN

Figure 3a,d shows the experimental adsorption isotherms according to the equilibrium Pb(Ⅱ) and Cr(VI) concentrations at room temperature, respectively. To gain some insight into the adsorption, the batch adsorption process was fitted using the two classical empirical models of Langmuir and Freundlich. Regression analysis of the linearized isotherms of Langmuir (*C_e_*/*q_e_* versus *C_e_*) and Freundlich (log *q_e_* versus log *C_e_*), using the slope and the intercept, gave the sorption constants(*q_m_*, *b* and *K_f_*, 1/*n*) and correlation coefficients (R^2^). The linear forms of the Langmuir and Freundlich equations are shown in Figure 3b,c,e,f. The *q_m_*, *b*, *K_f_*, 1/*n* values and the linear regression correlations for Langmuir and Freundlich modes are listed in Table 3. It could be observed that the AC based on WRN displayed good adsorption performance on heavy metal ions. The adsorption of Pb(Ⅱ) and Cr(VI) on the AC could follow both Freundlich and Langmuir type adsorption isotherms. However, based on the regression coefficients, the Langmuir isotherm fitted the experimental data slightly better than the Freundlich model, which indicates that there was a homogenous distribution of active sites on the surface of AC. Based on Langmuir isotherm, the AC based on WRN showed the highest metal uptake of 0.0583 mmol·g^−1^ (or 12.08 mg·g^−1^) for Pb(Ⅱ) and 0.180 mmol·g^−1^ (or 9.36 mg·g^−1^) for Cr(VI) (come from q_m_ data in Table 3). Compared with other AC adsorbents for the elimination of heavy metal ions (Table 4), the AC based on WRN show similar Cr(VI) adsorption capacity to some low-cost AC adsorbents produced from coconut shell [21] or peanut shell [23], as well as similar Pb(Ⅱ) adsorption capacity to the hazelnut-husk-based AC [25]. Combined with the low cost of WRN, the AC based on WRN had potential application in the treatment of heavy metal water pollution.

The AC showed a decreased BET surface areas after Cr(VI) and Pb(II) adsorption but maintained a similar type of isotherm and pore size distribution (see Table 1 and Appendix A). The slight reduction of carbon content from the ultimate analysis could be attributed to the incidental adsorption of heavy metal elements and other counter ions. As shown in Figure 4a,b, after the adsorption of heavy metals, the AC based on WRN could maintain its porous spherical structure, and heavy metal elements generated a homogenous distribution. The PXRD patterns of AC before and after Cr(VI) and Pb(II) adsorption revealed that the angle of graphite diffraction peaks of AC moved to lower angles after heavy metal adsorption (Figure 4c). Moreover, from the IR spectra of AC before and after heavy metal adsorption, the C–O vibration peak located at 1039 cm^−1^ red shift and the vibration peak of Pb–O and Cr–O were observed at 660 and 629 cm^−1^, respectively, which indicated that chemical action of AC and heavy metal ions occurred in the adsorption process (Figure 4d).

The AC based on WRN also shows good adsorption speed to heavy metal ions. As shown in Figure 5a,d, the AC could approach the adsorption equilibrium of both Pb(II) and Cr(VI) after 90 min adsorption in 25 mg·L^−1^ heavy metal solution. Furthermore, the pseudo-first-order and the pseudo-second-order kinetics models were employed to test of the experimental data of adsorption speed. The *q_e_*_1_, *q_e_*_2_, *K*_1_, *K*_2_ values were determined experimentally from the slope and intercept of straight-line adsorption kinetic plot. The linear forms of the pseudo-first-order and pseudo-second-order kinetic fitting are shown in Figure 5b,c,e,f. The values and regression coefficient are presented in Table 5. It could be observed that the adsorption process of Pb(II) and Cr(VI) by AC both accorded with the pseudo-second-order kinetic model better, which suggested that the heavy metal adsorption process of the AC was controlled by the chemisorption mechanism.

### 3.3. Structural Characterization of CQDs and CQDs/TiO_2_ Composite Based on WRN

As shown in Figure 6a–c, CQDs synthesized by WRN showed an oval or spherical morphology, whose (020) crystal plane with a lattice spacing of about 0.283 nm could be observed. Its particle size was mainly distributed between 1.5–2.5 nm, which was well dispersed and barely reunited. The IR spectrum indicated that there were carboxyl groups in the CQDs with an absorption peak at 1709 cm^−1^, which might play an important role in the combination of TiO_2_. The hydroxyl and alkyl groups also existed in the resulting CQDs. The vibration of the hydroxyl group was located at about 3424 cm^−1^. The stretch of C=C in aromatic rings was located at 1629 cm^−1^. The absorption peaks of 2926, 1384, and 669 cm^−1^ could be assigned to the stretching and bending vibration of alkyl groups (see Figure 6d).

As shown in Figure 7a, the CQDs/TiO_2_ composite presented a nearly square or egg-shaped nanostructure with a diameter range of 10–70 nm, and spherical particles of CQDs were evenly dispersed on the surface of TiO_2_. HRTEM image showed both the lattice interleaving of nano TiO_2_ and CQDs, in which the lattice stripes with a spacing of 0.149 nm and 0.114 nm belonged to (213) and (004) crystal planes of anatase TiO_2_, respectively, while the crystal plane spacing of 0.169 nm was highly matched with the (220) plane of rutile TiO_2_. In addition, a crystal plane with a lattice spacing of about 0.283 nm could be observed, which corresponded to the (020) crystal plane of CQDs (see Figure 7b). Thus, it was clear that the CQDs and nano TiO_2_ were successfully combined in the resulting composite. The formation of CQDs/TiO_2_ composite was further proved by IR characterization (see Figure 7c). Among them, the stretching vibration of the carboxyl group in original CQDs vanished after combination with TiO_2_, and the wide absorption band of pure TiO_2_ below 1000 cm^−1^ had an obvious red shift and narrowing after combination, which might be due to the reaction between carboxyl group in CQDs and hydroxyl group on the surface of TiO_2_. The XPS spectra showed the existence of C, O, and Ti elements in the resulting CQDs/TiO_2_ composite (see Figure 7d and Table 6). In the high-resolution spectrum of C 1 s, the characteristic peak at 284.71 eV could be attributed to C–C bond in CQDs, while the characteristic signals centered at 285.83 and 289.20 eV were related to the C–O bond and C=C bond, respectively (See Figure 7e). The high-resolution spectrum of O 1 s showed the existence of Ti–O, C–O, and O–H bonds, whose characteristic signals were located at 529.79, 531.30, and 532.64 eV, respectively. The characteristic peaks in Ti 2p spectrum at 458.50 and 464.20 eV belonged to the signals of Ti (2p_3/2_) and Ti (2p_1/2_). In addition, due to organic groups in CQDs, the point of zero charge (PZC) value of CQDs/TiO_2_ composite (6.77) is higher than that of pure TiO_2_ (6.03, see Figure 7h)

### 3.4. Photocatalytic Performance of CQDs/TiO_2_ Composite

The resulting CQDs/TiO_2_ composite displayed a high efficient photocatalytic degradation effect to various water-soluble dyes. The methylene blue [49,50] was one of the most common pollutants in dyeing waste water with considerable toxicity. As shown in Figure 8a, this methylene blue was difficult to degrade under 405 nm visible purple light. After 80 min of illumination, the degradation rate was only 3.32%. The pure nano TiO_2_ had a catalytic effect on methylene blue but a complete degradation could not be achieved, with a degradation rate of 3.18% within 10 min and 59.85% within 80 min. By contrast, the CQDs/TiO_2_ composite based on WRN had much better photocatalytic performance, whose degradation rate could be as high as 81.84% within 10 min and 99.87% within 80 min. Moreover, the CQDs/TiO_2_ composite could be recycled and reused without any appreciable decrease in degradation rate, which could maintain a degradation rate of more than 96.5% after seven photocatalytic cycles (See Figure 8b). In addition, the degradation rate of CQDs/TiO_2_ composite could be affected by the loading dosage of CQD_S_. For example, the degradation rate of the CQD_S_/TiO_2_ sample with a carbon content of 4.96 wt% (CQD_S_/TiO_2_–4) showed a degradation rate of 93.3% within 80 min, which was slightly lower than other CQDs/TiO_2_ composite with higher carbon content (see Appendix A).

Under natural light (simulated by a xenon lamp), CQDs/TiO_2_ based on WRN also showed better photocatalytic performance than that of pure TiO_2_. The degradation rate of pure TiO_2_ within 10 and 80 min was 46.71 and 49.12%, respectively, which could not achieve complete degradation. By contrast, the degradation rate of CQDs/TiO_2_ could be up to 95.95% within 10 min and 99.54% within 80 min, which was even better than that under 405 nm purple light (see Figure 8c).

As shown in Figure 9, the resulting CQDs/TiO_2_ composite displayed a high efficient photocatalytic degradation effect on other common water-soluble dyes such as malachite green, methyl violet, basic fuchsin, and rhodamine B under visible light irradiation and showed better photocatalytic performance than that of commercial TiO_2_. Among them, the degradation rate of CQDs/TiO_2_ composite of malachite green and methyl violet could reach up to 99% within 10 min, which indicated that this composite displayed a very high photocatalytic degradation effect on triphenylmethane dye (See Figure 9a,b). It was also observed that the CQDs/TiO_2_ composite could be used against some dyes with good photobleaching resistance. For example, rhodamine B was almost undegradable under 405 nm purple light with the existence of pure TiO_2_, but the CQDs/TiO_2_ composite could degrade 99% rhodamine B within 30 min (see Figure 9d). Furthermore, due to the very low cost of CQDs based on WRN, the CQDs/TiO_2_ composite based on WRN could also achieve a lower cost than that of commercial TiO_2_, which could be a good substitute for commercial TiO_2_ and had good application in organic water pollution control. In addition, the photocatalytic degradation processes of the dyes by CQDs/TiO_2_ composite all accorded with the Langmuir Hinshelwood kinetic model, whose apparent constant (K_app_) values and regression coefficients are presented in Table 7. The K_app_ values displayed a range of methyl violet > malachite green > rhodamine B > methylene blue > basic fuchsin.

In order to further export the photocatalytic mechanism of the as-prepared CQDs/TiO_2_ composite based on WRN, the UV-VIS absorption spectra of the pure TiO_2_ and CQDs/TiO_2_ composite were tested and shown in Figure 10a. The absorption of the pure TiO_2_ was mainly in the UV region, and there was almost no absorption in the wavelength range above 430 nm. In contrast, the CQDs/TiO_2_ composite exhibited a wide visible-light absorption region and enhanced light-absorption intensity. Thus, the combination of CQDs and TiO_2_ could effectively enhance the energy absorption and produce more electron-hole pairs in the visible light region. Furthermore, the band gaps of the pure TiO_2_ and CQDs/TiO_2_ samples were calculated by means of Kubelka–Munk theory [51] and shown in Figure 10b. The band gap of the pure TiO_2_ photocatalyst was about 3.07 eV. By comparison, the introduction of CQDs could afford a narrower band gap of 1.55 eV, which could promote the electronic transition and improve the photocatalytic degradation process. This feature of the CQDs/TiO_2_ composite was also proven by photocurrent response with on/off cycles of visible light irradiation (Figure 10c). It could be observed that the CQDs/TiO_2_ electrode displayed a much higher photocurrent (more than 60 times) in comparison to the pure TiO_2_ electrode upon irradiation. This means that a faster interfacial charge transfer to the electron acceptor occured in the CQDs/TiO_2_ composite, which resulted in the more effective separation of electron-hole pairs. On the other hand, the steady-state PL spectra of the pure TiO_2_ and CQDs/TiO_2_ composite (Figure 10d) showed that the emission intensity of the CQDs/TiO_2_ composite was much lower than that of pure TiO_2_ at similar emission, which suggested that the modification of CQDs could effective reduce the recombination of electron-hole pairs.

The ESR spin-trap with the DMPO technique was used to investigate the reactive oxygen species generated by the CQDs/TiO_2_ composite or pure TiO_2_ photocatalysts. As shown in Figure 11a,b, obvious characteristic peaks of both superoxide radical (O_2_^·−^) and hydroxyl radical (·OH) of DMPO was observed by using the CQDs/TiO_2_ composite as a photocatalyst, which suggested that the CQDs/TiO_2_ composite could reduce adsorbed O_2_ to form O_2_^·−^ radical and oxidize adsorbed H_2_O or OH^−^ to form ·OH radical under light radiation. Moreover, compared with using pure TiO_2_ as a photocatalyst, signals of both O_2_^·−^ and ·OH radical of DMPO with CQDs/TiO_2_ composite were evidently stronger (Figure 11c,d). Thus the CQDs/TiO_2_ composite displayed stronger abilities in producing O_2_^·−^ and ·OH radical than that of pure TiO_2_ on the same test condition, which demonstrates that the introduction of CQDs could result in efficient electron-hole pair separation and enable more photogenerated electrons to reduce O_2_ and more photogenerated holes to oxidize H_2_O or OH^−^. Based on the results of photodegradation and reactive oxygen species during the degradation process, possible photocatalytic mechanism schematics was illustrated in Figure 11e. When the CQDs/TiO_2_ composite was irradiated by visible light, the CQDs can be easily excited with photogenerated electrons on the conduction band (CB) and leave the holes on the valence band (VB). The photogenerated hole could react with H_2_O or OH^−^ to produce more ·OH. The excited electrons could spatial transfer to the TiO_2_ quickly and result in a more efficient electron-hole pair separation. After that, the separated electrons on the CQDs could react with O_2_ to produce more O_2_^·−^. The generated O_2_^·−^ and ·OH radical play the main role in the photodegradation process, causing excellent photocatalytic activity.

## 4. Conclusions

In this paper, low-cost AC and CQDs/TiO_2_ composite were simultaneously synthesized by using WRN as raw material. During the synthesis process, the intermediate mixture after hydrothermal carbonization was divided into two parts: the precipitate (HTC powder) was used to produce AC, while the liquid part (CQDs solution) could acquire synthesized CQDs/TiO_2_ composite simultaneously. Both of the two materials showed potential applications in water pollution control. The AC could absorb heavy metal ions efficiently, while the CQDs/TiO_2_ composite displayed a high efficient photocatalytic degradation effect on various water-soluble dyes, which showed better photocatalytic performance than commercial TiO_2_, which could form a low-cost waste-water treatment system together with AC. Compared with other recycling methods for catering waste, the method described in this paper could achieve the full utilization of WRN without secondary pollution and could produce about 43 g AC and 29 g CQDs/TiO_2_ composite for each kilogram of WRN. On the other hand, the AC or CQDs/TiO_2_ composite based on WRN could realize higher added value than conventional products like feed, a fact that suggests broad prospects for industrialization and could provide a new method for the recycling of cooking waste.

## Figures and Tables

**Figure 1 nanomaterials-12-00472-f001:**
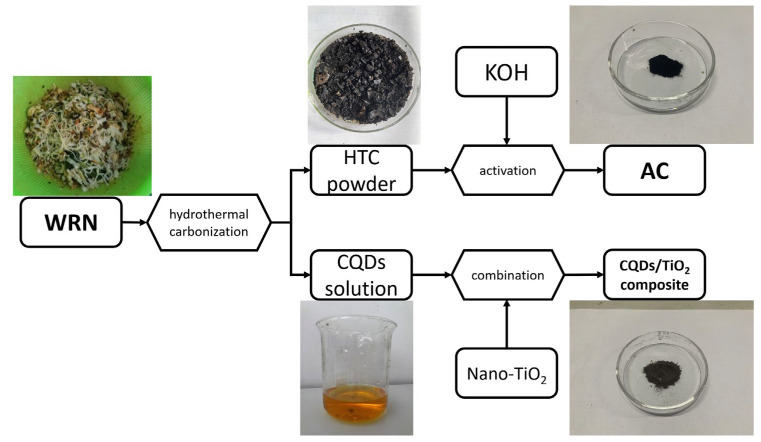
Schematic diagram of the simultaneous synthesis of AC and CQDs/TiO_2_ composite.

**Figure 2 nanomaterials-12-00472-f002:**
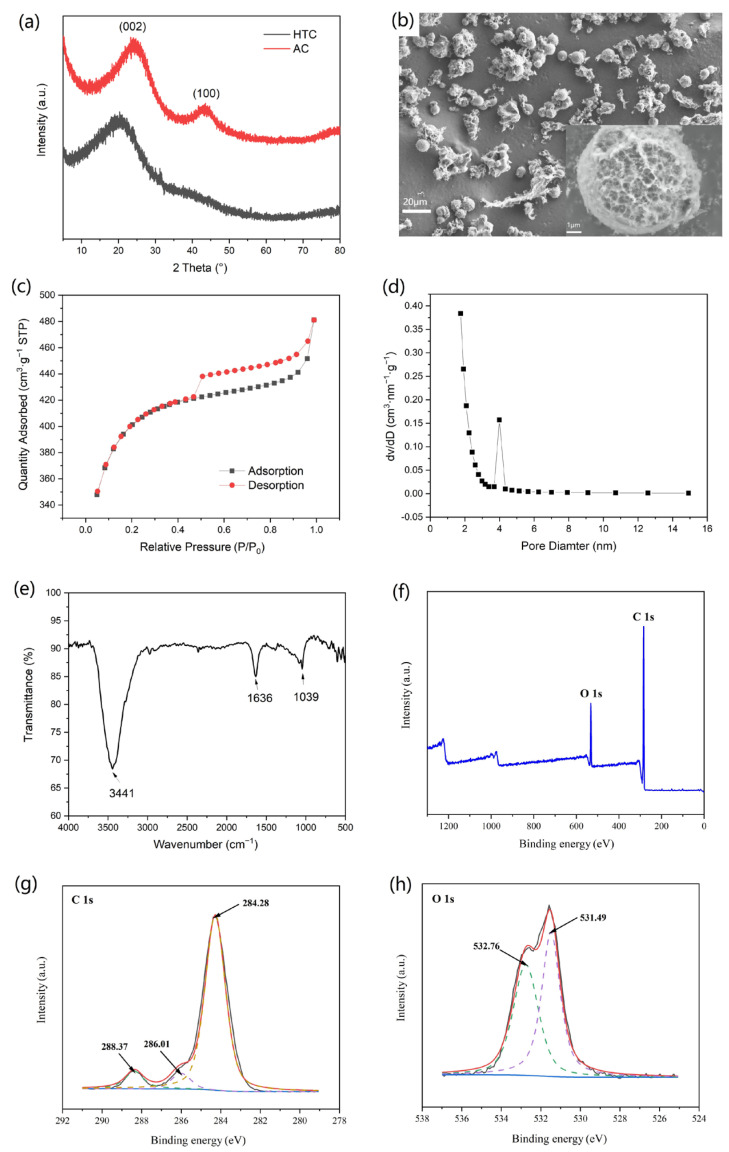
(**a**–**c**): The PXRD pattern (**a**), SEM image (**b**), nitrogen adsorption/desorption isotherms (**c**), pore-size-distribution curves (**d**), and IR spectrum (**e**) of AC based on WRN; (**f**–**h**): The full XPS (**f**), C 1 s (**g**), and O 1 s (**h**) high-resolution spectra of AC based on WRN.

**Figure 3 nanomaterials-12-00472-f003:**
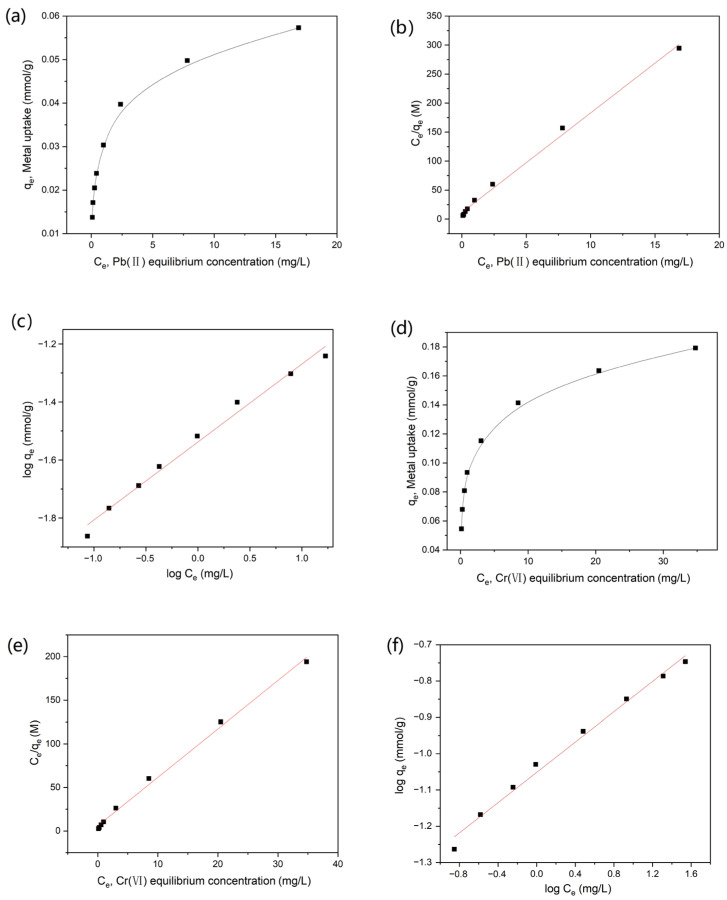
(**a**–**c**): The adsorption isotherm of resulted in AC to Pb(II) by equilibrium concentration (**a**), Langmuir equation (linear form, (**b**)) and Freundlich equation (linear form, (**c**)); (**d**–**f**): the adsorption isotherm of resulted from AC to Cr(VI) by equilibrium concentration (**d**), Langmuir equation (linear form, (**e**)), and Freundlich equation (linear form, (**f**)).

**Figure 4 nanomaterials-12-00472-f004:**
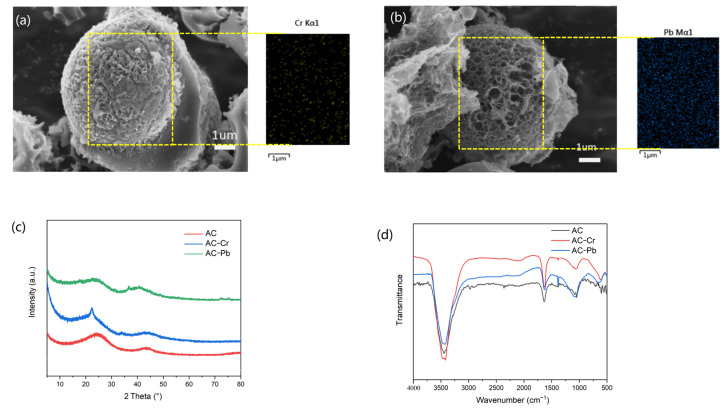
(**a**,**b**) The SEM image and heavy metal distribution image of the AC sample after Cr(VI) (**a**) and Pb(II) (**b**) adsorption; (**c**) the PXRD patterns of AC before and after Cr(VI) and Pb(II) adsorption; (**d**) the IR spectra of AC before and after Cr(VI) and Pb(II) adsorption. The adsorption samples were obtained in 100 mg·L^−1^ heavy metal solution with an adsorbent dosage of 7 g·L^−1^ for 12 h.

**Figure 5 nanomaterials-12-00472-f005:**
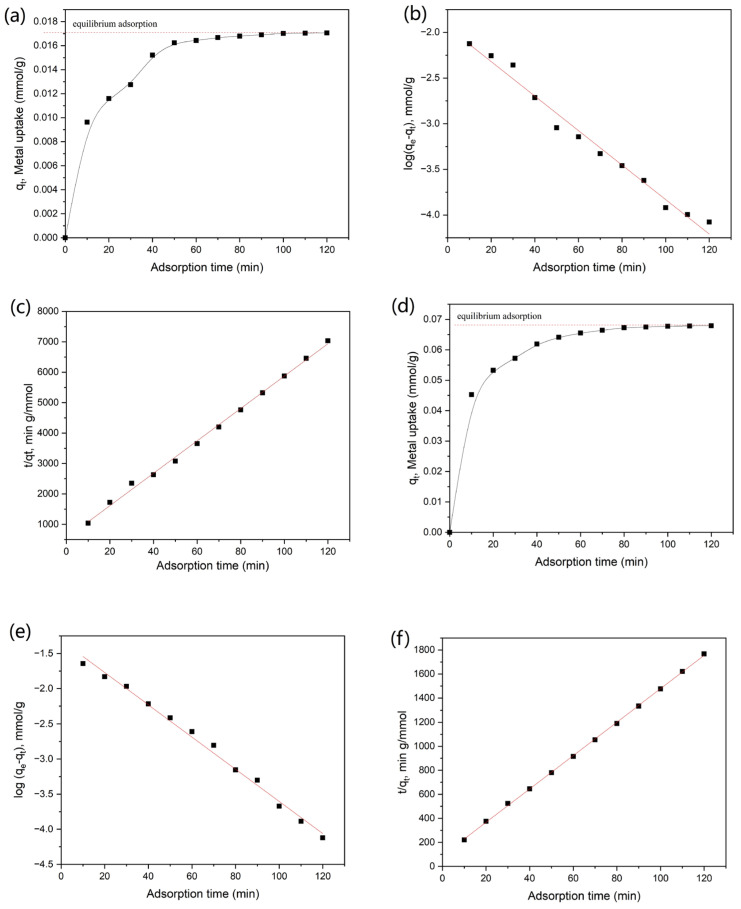
(**a**–**c**) Adsorption of Pb(II) with AC by contact time (**a**), pseudo-first-order kinetics (**b**) and pseudo-second-order (**c**) kinetics mode; (**d**–**f**) adsorption of Cr(VI) with AC by contact time (**d**), pseudo-first-order kinetics (**e**) and pseudo-second-order (**f**) kinetics mode.

**Figure 6 nanomaterials-12-00472-f006:**
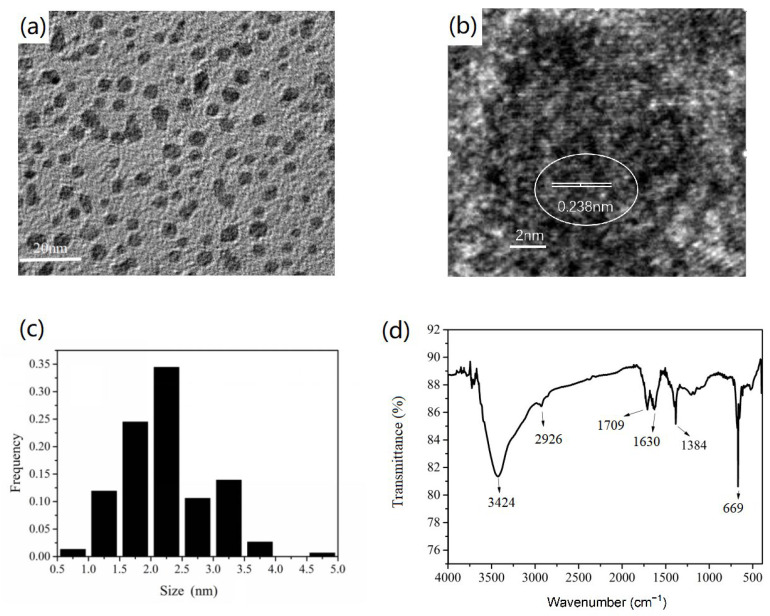
The TEM image (**a**), HRTEM image (**b**), particle-size distribution (**c**), and IR spectrum (**d**) of CQDs based on WRN.

**Figure 7 nanomaterials-12-00472-f007:**
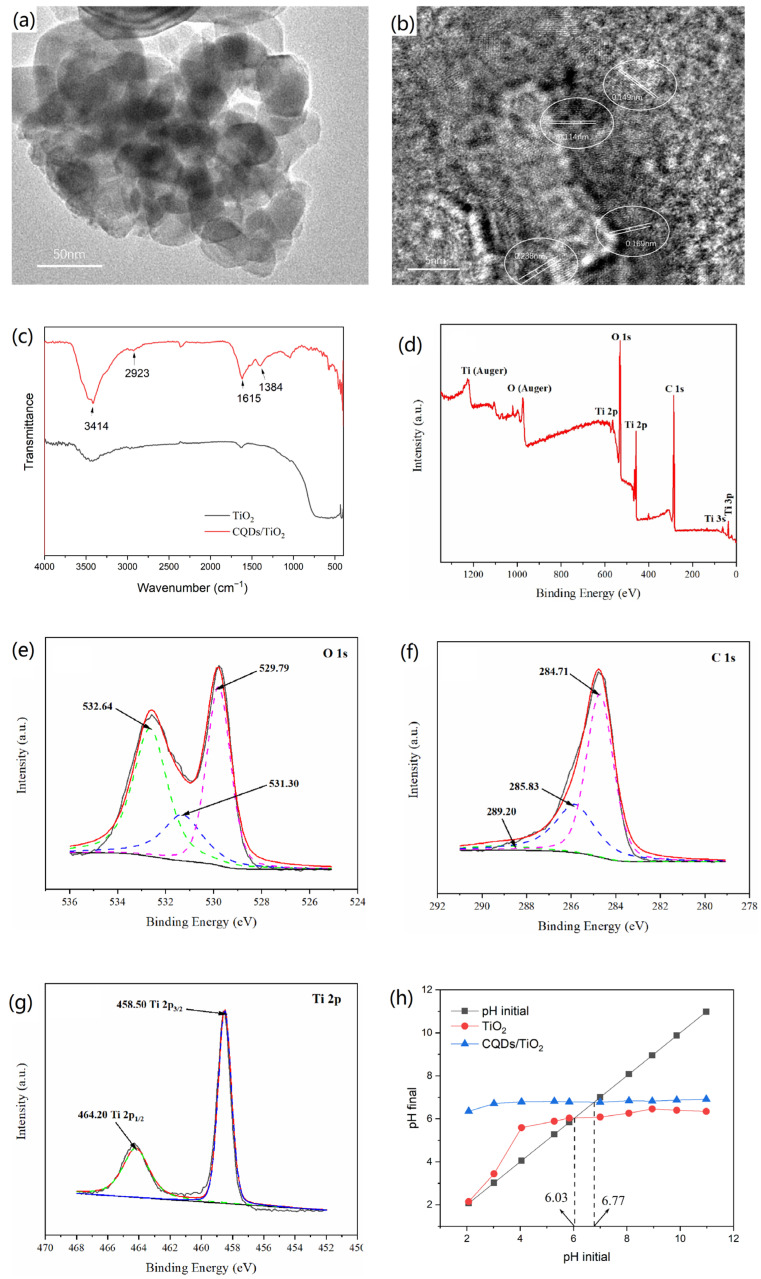
(**a**,**b**) The TEM image (**a**) and HRTEM image (**b**) of CQDs/TiO_2_ composite based on WRN; (**c**) IR spectra pure TiO_2_ and CQDs/TiO_2_ composite. (**d**–**g**): The full XPS (**d**), O 1 s (**e**), C 1 s (**f**), and Ti 2p (**g**) high-resolution spectra of CQDs/TiO_2_ composite based on WRN. (**h**) PZC of pure TiO_2_ and CQDs/TiO_2_ composite.

**Figure 8 nanomaterials-12-00472-f008:**
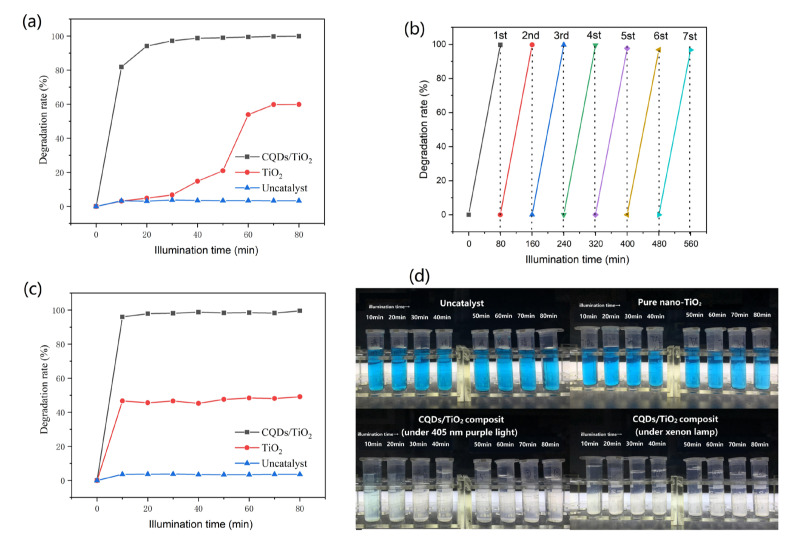
(**a**) The photocatalytic degradation rate of CQDs/TiO_2_ composite and pure TiO_2_ to methylene blue within different irradiation times under 405 nm purple light; (**b**) the photocatalytic degradation rate of CQDs/TiO_2_ composite of the different photocatalytic cycle with 80 min as a working cycle; (**c**) the photocatalytic degradation rate of CQDs/TiO_2_ composite and pure TiO_2_ methylene blue within different irradiation time under a xenon lamp; (**d**) the photographs showing the photocatalytic degradation effects on methylene blue of CQDs/TiO_2_ composite and pure TiO_2_.

**Figure 9 nanomaterials-12-00472-f009:**
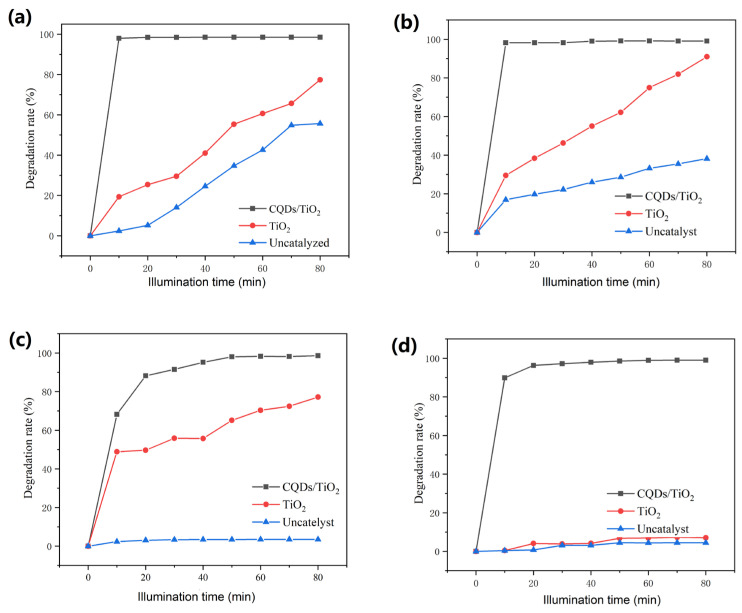
The photocatalytic degradation rate of CQDs/TiO_2_ composite and pure TiO_2_ to malachite green (**a**) methyl violet (**b**), basic fuchsin (**c**), and rhodamine B (**d**) within different irradiation times under 405 nm purple light.

**Figure 10 nanomaterials-12-00472-f010:**
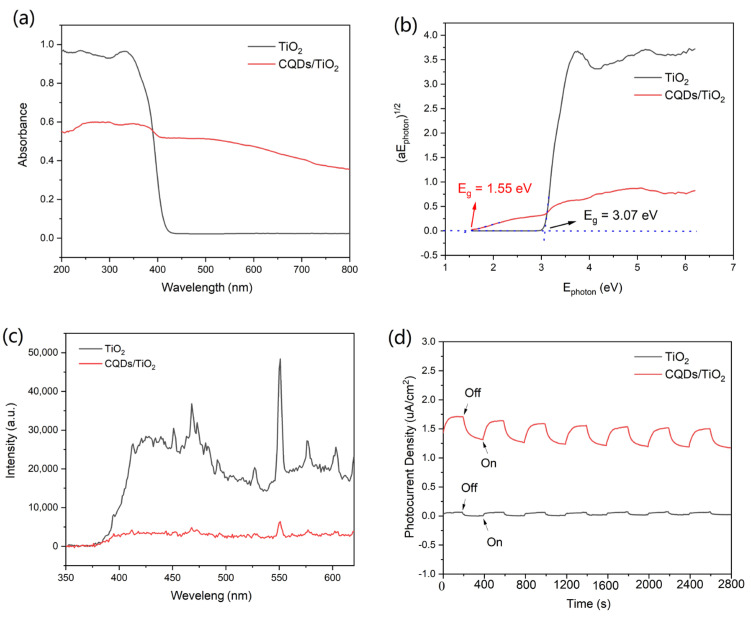
(**a**) The UV-VIS absorption spectra of CQDs/TiO_2_ composite and pure TiO_2_; (**b**) the (aE_photon_)_1/2_ vs. E_photon_ curves of CQDs/TiO_2_ composite and pure TiO_2_; (**c**) the PL spectra of CQDs//TiO_2_ composite and pure TiO_2_; (**d**) the photocurrent response of CQDs/TiO_2_ composite and pure TiO_2_.

**Figure 11 nanomaterials-12-00472-f011:**
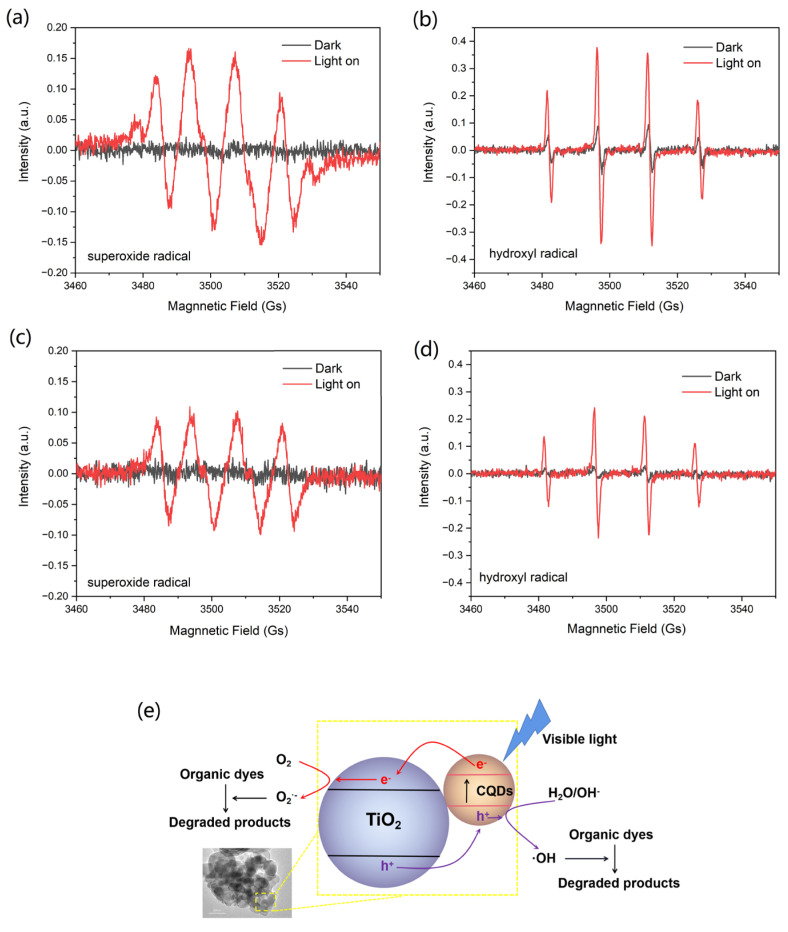
(**a**,**b**) DMPO spin-trapping ESR spectra of CQDs/TiO_2_ composite in methanol (**a**) and water (**b**) under visible light irradiation; (**c**,**d**) DMPO spin-trapping ESR spectra of pure TiO_2_ in methanol (**c**) and water (**d**) under visible light irradiation; (**e**) schematic illustration of the photocatalytic mechanism CQDs/TiO_2_ composite.

**Table 1 nanomaterials-12-00472-t001:** Ultimate analyses and characteristic of AC, HTC, as well as AC after Pb(II) and Cr(VI) adsorption.

Product	Ultimate Analysis (wt. %)			BET Surface Area (m^2^·g^−1^)	Micropore Volume (cm^3^·g^−1^)	Average Pore Width (nm)	Point of Zero Charge (PZC)
	C	H	N				
AC	84.17	1.21	1.30	1250.35	0.4053	2.4773	5.83
HTC	51.03	4.20	1.58	0.18	0.0004	-	5.04
AC–Cr ^a^	82.95	1.30	1.28	793.12	0.3136	2.3959	5.97
AC–Pb ^b^	80.99	1.26	1.84	803.25	0.2983	2.3647	5.64

^a^ The AC sample after Cr(VI) adsorption in 100 mg/L Cr(VI) solution with an adsorbent dosage of 7 g·L^−1^ for 12 h; ^b^ The AC sample after Pb(II) adsorption in 100 mg/L Pb(II) solution with an adsorbent dosage of 7 g·L^−1^ for 12 h.

**Table 2 nanomaterials-12-00472-t002:** The assignment of the XPS peaks of AC based on WRN.

Element	Peak (eV)	Surface Group	Assignment
C1s	284.28	C	Graphitic carbon
	286.01	C–O	Phenolic, alcoholic or etheric structure
	288.37	C=C	π-electrons in aromatic ring
O1s	531.49	C–O	Oxygen singly bonded to carbon in aromatic rings
	532.76	O–H	Hydroxyl group

**Table 3 nanomaterials-12-00472-t003:** Adsorption parameters of the Langmuir and Freundlich isotherms at room temperature for the adsorption of Pb(II) and Cr(VI).

Heavy Metal Ion	Langmuir			Freundlich		
	*q_m_* (mmol·g^−1^)	*b* (L·mg^−1^)	R^2^	*K_f_* (mmol·g^−1^)	1/*n*	R^2^
Pb(II)	0.0583 ± 0.0016	0.8614 ± 0.0049	0.9954	0.0294 ± 0.0006	0.2687 ± 0.0127	0.9867
Cr(VI)	0.1800 ± 0.0048	1.0415 ± 0.0088	0.9956	0.0134 ± 0.0014	0.2085 ± 0.0085	0.9899

**Table 4 nanomaterials-12-00472-t004:** Adsorption capacity of Pb(II) and Cr(VI) by AC with different low cost raw materials.

Heavy Metal Ion	Raw Materials	Adsorbent Dosage (g·L^−1^)	pH	Adsorption Capacity (mg·g^−1^)	Reference
Pb(II)	Polygonum orientale Linn.	0.6	5	98.39	[28]
	Pecan shell	0.5–10	4.8	64.20	[22]
	Date pit	4.0	5.2	30.70	[24]
	Coconut shell	2	5	21.88	[20]
	Hazelnut husk	12	5.7	13.05	[25]
	Waste rice noodles (WRN)	7	5	12.08	This work
Cr(VI)	Paper mill sludge	1.5–7	4	23.18	[27]
	Peanut shell	1	2	8.31	[23]
	Eichhornia crassipes root	7	4.5	36.34	[29]
	Coconut shell	2	4	10.88	[21]
	Sagowaste	2	2	5.78	[26]
	Waste rice noodles (WRN)	7	5	9.36	This work

**Table 5 nanomaterials-12-00472-t005:** The kinetic adsorption parameters were obtained using pseudo-first-order and pseudo-second-order for the adsorption of Cr(VI) and Pb(II).

Heavy Metal Ion	Pseudo-First-Order			Pseudo-Second-Order		
	*K*_1_ (min^−1^)	*q_e_*_1_ (mmol·g^−1^)	R^2^	*K*_2_ (g·mmol·min^−1^)	*q_e_*_2_ (mmol·g^−1^)	R^2^
Pb(II)	0.0435 ± 0.0002	0.0114 ± 0.0001	0.9832	5.1481 ± 0.0582	0.0188 ± 0.0003 ^a^	0.9974
Cr(VI)	0.0527 ± 0.0001	0.0488 ± 0.0005	0.9924	2.1421 ± 0.0138	0.0720 ± 0.0004 ^b^	0.9997

^a^ The experimental value (*q_e_*) of Pb(II) was 0.0172 mmol·g^−1^; ^b^ The experimental value (*q_e_*) of Cr(VI) was 0.0680 mmol·g^−1^.

**Table 6 nanomaterials-12-00472-t006:** The assignment of the XPS peaks of CQDs/TiO_2_ composite based on WRN.

Element	Peak (eV)	Surface Group	Assignment
C 1 s	284.71	C	Graphitic carbon
	285.83	C–O	Phenolic, alcoholic or etheric structure
	289.20	C=C	π-electrons in aromatic ring
O 1 s	529.79	Ti–O	Oxygen bonded to titanium
	531.30	C–O	Oxygen singly bonded to carbon in aromatic rings
	532.64	O–H	Hydroxyl group
Ti 2p	458.50	Ti	Ti (2p_3/2_)
	464.20	Ti	Ti (2p_1/2_)

**Table 7 nanomaterials-12-00472-t007:** The kinetic adsorption parameters obtained using pseudo-first-order for the photocatalytic degradation of various dyes under 405 nm purple light.

Dyes	K_app_ (min^−1^)	R^2^
methylene blue	0.1238 ± 0.0237	0.9967
malachite green	0.2586 ± 0.0104	0.9935
methyl violet	0.3032 ± 0.0082	0.9971
basic fuchsin	0.1199 ± 0.0032	0.9971
rhodamine B	0.1949 ± 0.0067	0.9953

## Data Availability

Not applicable.

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
