# Peer review of "Activated Carbon and Carbon Quantum Dots/Titanium Dioxide Composite Based on Waste Rice Noodles: Simultaneous Synthesis and Application in Water Pollution Control"

_nanomaterials, 2022, doi:10.3390/nano12030472_

Round 1

Reviewer 1 Report

In the manuscript entitled „Activated carbon and carbon quantum dots/titanium dioxide composite based on waste rice noodles: Simultaneous synthesis and application in water pollution control“ the authors synthesized activated carbon (AC) and carbon quantum dots/titanium dioxide (CQDs/TiO2) composite by using waste rice noodle (WRN) as starting precursors. My major remarks are the following:

  1. In the introduction section state of the art related to photocatalysis of CQDs should be inserted. The following references should be inserted as well: J. Prekodravac et al., Journal of Photochemistry & Photobiology, B: Biology 200 (2019) 111647.
  2. More experimental details related to TEM, XRD, XPS should be given.
  3. The mechanism of photocatalytic activity of CQDs should be explained.
  4. The production of hydroxyl radicals must be checked by PL oe EPR method.
  5. The point zero measurements must be conducted.

Author Response

Dear Reviewer,

I am writing regarding to our manuscript (Title: Activated carbon and carbon quantum dots/titanium dioxide composite based on waste rice noodles: Simultaneous synthesis and application in water pollution control; No. nanomaterials-1544524), which is currently under review by Nanomaterials. We appreciate that you gave us a chance of revision to improve our manuscript to a level suitable for publication in your journal. We would like to thank for the precious comments from you.

Based on your comment and suggestion, the manuscript has been revised. The modified parts are shown using the track changes mode. The responses to your comment are listed below.

Comments

  • In the introduction section state of the art related to photocatalysis of CQDs should be inserted. The following references should be inserted as well: J. Prekodravac et al., Journal of Photochemistry & Photobiology, B: Biology 200 (2019) 111647.

Our responses: Based on your suggestion, the introduction section was revised and expanded. The importance and significance of CQDs to photocatalysts were described in the revised manuscript. More than 10 literature (including the work: J. Prekodravac et al., Journal of Photochemistry & Photobiology, B: Biology 2019, 200, 111647) were added to the revised manuscript.

  • More experimental details related to TEM, XRD, XPS should be given.

Our responses: Based on your suggestion, the experimental details related to BET, TEM, XRD, XPS and other characterizations were added in the revised manuscript.

  • The mechanism of photocatalytic activity of CQDs should be explained.

Our responses: Based on your suggestion, the photocatalytic mechanism CQDs/TiO2 composite was investigated by UV-VIS absorption spectra, PL spectra, photocurrent response and spin-trapping ESR spectra. The results and discussions of the photocatalytic mechanism CQDs/TiO2 composite were added in the revised manuscript.

  • The production of hydroxyl radicals must be checked by the PL oe EPR method.

Our responses: Based on your suggestion, the production abilities of superoxide radical and hydroxyl radicals of both CQDs/TiO2 composite and pure TiO2 were investigated by the EPR method, whose results and discussions were added in the revised manuscript.

  • The point zero measurements must be conducted.

Our responses: Based on your suggestion, the measurements of point of zero charge (PZC) to HTC, AC, CQDs/TiO2 composite and pure TiO2 were carried out, whose results and discussions were added in the revised manuscript.

Thank you very much for your consideration!

Sincerely Yours,

Dr. Shuo-ping Chen

Reviewer 2 Report

Reviewers Comments

Manuscript ID:  nanomaterials-1544524

Title Activated carbon and carbon quantum dots/titanium dioxide composite based on waste rice noodles: Simultaneous synthesis and application in water pollution control

Journal: nanomaterials

General comments

The study reports the synthesis and full characterization of activated carbon (AC) and carbon quantum dots/titanium dioxide (CQDs/TiO2) composite using waste rice noodle (WRN for adsorptive removal of Pb(â…¡) and Cr(â…¥) and photocatalytic degradation removal of dyes such as methylene blue, malachite green, methyl violet, basic fuchsin and rhodamine B were investigated as well.

The manuscript is quite interesting, and 20% similarity index I suppose is acceptable, but need to be confirmed from the journal. Though the manuscript was well written and presents significant characterizations and experimental results analysis, yet it lacks significant aspects related to adsorption and photocatalytic mechanisms presentation and interpretations. A major revision is needed prior to the acceptance of manuscript as per the following comments provided below 

Specific comments

  1. The maximum adsorption capacity of the new adsorbents as well as the major sorption and photocatalytic mechanisms are missing in the abstract.
  2. Introduction part not sufficient information is provided. A more comprehensive review on the subject matter with relevant and most updated references is needed.
  3. Adsorption equilibrium and thermodynamic studies are missing. Adsorption mechanism cannot be understood without equilibrium and thermodynamic data. As such authors should add equilibrium and thermodynamic studies to complement present data, discussions, and mechanisms elucidation to help fully understanding the potentials of the new adsorbent in pollution
  4. Correct mechanisms for the metals adsorption should be properly clarified and established using the full characterization of the final adsorbent using techniques such as BET, SEM, EDX, FTIR, TGA, and XRD, elemental compositions analyses and surface charges (zeta potential, point of zero charge) results for both before and after adsorption (each one superimposed)
  5. Provide and discuss the detailed and comparative for the precursor materials and the final adsorbent before and after adsorption to further support understanding the adsorption mechanism.
  6. There are many other similar adsorbents and catalysts based novel adsorbent composites in literature that might have performed better than the reported adsorbent. Provide a table format comparison with the respective adsorbents’ dosages, contact time and sample volumes used etc.
  7. Kinetic as well as other required models’ fittings parameters; R2, models error and well as RMSE should be provided and serve as a guide for selecting the best model for better mechanisms interpretations

Author Response

Dear Reviewer,

I am writing regarding to our manuscript (Title: Activated carbon and carbon quantum dots/titanium dioxide composite based on waste rice noodles: Simultaneous synthesis and application in water pollution control; No. nanomaterials-1544524), which is currently under review by Nanomaterials. We appreciate that you gave us a chance of revision to improve our manuscript to a level suitable for publication in your journal. We would like to thank you for the precious comments from you.

Based on your comment and suggestion, the manuscript has been revised. The modified parts are showed using the track changes mode. The responses to your comment are listed in below.

Comments

  • The maximum adsorption capacity of the new adsorbents as well as the major sorption and photocatalytic mechanisms are missing in the abstract.

Our responses: Based on your suggestion, the maximum adsorption capacity of AC, as well as the major adsorption and photocatalytic mechanisms were added in the abstract of a revised manuscript.

  • Introduction part not sufficient information is provided. A more comprehensive review on the subject matter with relevant and most updated references is needed.

Our responses: Based on your suggestion, the introduction section was revised and expanded. The importance and significance of AC wastewater treatment, as well as the CQDs to photocatalysts was described in the revised manuscript. More than 10 literature were added to the revised manuscript.

  • Introduction part not sufficient information is provided. A more comprehensive review on the subject matter with relevant and most updated references is needed.

Our responses: Based on the your suggestion, the introduction section was revised and expanded. The importance and significance of AC wastewater treatment, as well as the CQDs to photocatalysts was descried in the revised manuscript. More than 10 literatures were added to the revised manuscript.

  • Correct mechanisms for the metals adsorption should be properly clarified and established using the full characterization of the final adsorbent using techniques such as BET, SEM, EDX, FTIR, TGA, and XRD, elemental compositions analyses and surface charges (zeta potential, point of zero charge) results for both before and after adsorption (each one superimposed)

Our responses: Based on the your suggestion, the characterization of AC before and after Cr(â…¥) and Pb(â…¡) adsorption were carried out with BET, SEM, EDX, FTIR, XRD, EA and PZC test, whose results and discussions were added in section 3.2 of the revised manuscript.

  • Provide and discuss the detailed and comparative for the precursor materials and the final adsorbent before and after adsorption to further support understanding the adsorption mechanism.

Our responses: Based on your suggestion, the investigation of precursor materials (HTC powder) was carried out, while the comparisons between HTC powder and AC product were also added in the revised manuscript.

  • There are many other similar adsorbents and catalysts based novel adsorbent composites inthe literature that might have performed better than the reported adsorbent. Provide a table format comparison with the respective adsorbents’ dosages, contact time and sample volumes used etc.

Our responses: Based on your suggestion, the comparison of adsorption capacities of the resulted AC and other reported adsorbents was added as Table 4 in the revised manuscript.

  • Kinetic as well as other required models’ fittings parameters; R2, models error and well as RMSE should be provided and serve as a guide for selecting the best model for better mechanisms interpretations

Our responses: Based on your suggestion, all fittings parameters and regression correlations were listed as Table 3, Table 5 and Table 7 in the revised manuscript.

Thank you very much for your consideration!

Sincerely Yours,

Dr. Shuo-ping Chen

Round 2

Reviewer 1 Report

I do not have any comments.

Reviewer 2 Report

The authors have addressed all comments and manuscript can be published as is